# Sero-Epidemiology of *Coxiella burnetii* Infection in Small Ruminants in the Eastern Region of Punjab, Pakistan

**DOI:** 10.3390/pathogens11060664

**Published:** 2022-06-08

**Authors:** Freeha Amin, Shahzad Ali, Arshad Javid, Muhammad Imran, Muhammad Imran Rashid, Katja Mertens-Scholz, Heinrich Neubauer

**Affiliations:** 1Department of Wildlife & Ecology, University of Veterinary and Animal Sciences, Ravi Campus, Pattoki 55300, Pakistan; fareehaamin93@gmail.com (F.A.); arshadjavid@uvas.edu.pk (A.J.); 2Institute of Biochemistry and Biotechnology, University of Veterinary and Animal Sciences (UVAS), Lahore 54000, Pakistan; muhammad.imran@uvas.edu.pk; 3Department of Parasitology, University of Veterinary and Animal Sciences (UVAS), Lahore 54000, Pakistan; imran.rashid@uvas.edu.pk; 4Friedrich-Loeffler-Institut, Institute of Bacterial Infections and Zoonoses, 07743 Jena, Germany; katja.mertens-scholz@fli.de (K.M.-S.); heinrich.neubauer@fli.de (H.N.)

**Keywords:** *Coxiella burnetii*, livestock, serology, risk factors

## Abstract

The aim of this study was to investigate the seroprevalence of Q fever in sheep and goats in Kasur, Okara, and Pakpattan in the Punjab of Pakistan. Q fever is a widely reported zoonotic disease caused by *Coxiella* (*C*.) *burnetii*. The main reservoirs are small ruminants that excrete the bacteria in birth by-products in high numbers. Thus, the bacteria can also be detected in the air and the dust of livestock farms. The infection is often asymptomatic in ruminants, but it can lead to reproductive disorders. This cross-sectional study found that a significant number (n = 43; 11.3%) of 300 randomly selected small ruminants of nine tehsils were seropositive using a commercially available ELISA. Seroprevalence was significantly higher in goats (17.1%) than in sheep (4.9%). Binary logistic regression analysis proved that species, age, and breed have a significant effect on the prevalence of Q fever. Tick infestation, contact with animal fomites, contact with other animals, production system, and health status of an animal had a significant impact on the prevalence of Q fever. These findings on Q fever in animals can be used to improve the visibility of this zoonotic disease. These findings will help local health authorities to focus on the origin of the problem and facilitate applying preventive measures to the affected livestock farms.

## 1. Introduction

One of the most common zoonotic diseases at the global level is Q fever which is caused by γ-Proteobacteria *Coxiella* (*C*.) *burnetii*. Q fever, generally known as Query fever, is an airborne zoonotic infection with an impact on public health and initially unknown (query) cause. It is a notorious zoonotic agent due to its high physical resistance against a tough environment and its high infectivity [1]. This resistance is based on some of its cell forms. Once the pathogen has entered cells, it undergoes a morphological differentiation from the spore-like small cell variant (SCV) to the metabolic active and replicative large cell variant [2]. The contagious nature of *C. burnetii* also favors aerosol-borne infections as less than 10 bacteria are sufficient to cause disease. Placental tissue can indeed contain more than 10^9^ bacteria per gram. Thus, foetal membranes, amniotic fluids, vaginal discharge, and the placenta pose a high risk for contamination of the surrounding environment during parturition or at the time of abortion, respectively [3,4]. Additionally, the bacteria can be shed via milk, feces, and urine for months. Thus, the transmission of *C. burnetii* is favored by the landscape and climatic conditions of a specific region where it can easily be transported with the wind over long distances of up to 10 miles [5]. These characteristics explain why the disease is more prevalent in areas with dusty and warm climates, e.g., in the east of Turkey [6,7]. Q fever is registered in the list of emerging diseases by the WHO and US CDC [3,8,9].

Q fever in humans leads to clinical disease in 40% of infected persons presenting with flu-like symptoms such as headache and fever and atypical pneumonia or granulomatous hepatitis in 10% of patients. In 1 to 2% of patients, the disease becomes chronic with endocarditis and chronic fatigue syndrome may be seen. In animals, Q fever does not show a pathognostic syndrome but has a high impact on livestock and wildlife [2,10]. Placentitis, abortions, pneumonia, conjunctivitis, and hepatitis may be seen. The only symptom which often indicates the presence of Q fever in livestock is abortion, especially in the late stages [11,12]. Q fever can be a reason for the loss of livestock and thus, losses to a country’s economy. 

Several wild and domesticated mammal species can be reservoirs of *C. burnetii* and may transmit the infection to other livestock species and humans. *C. burnetii* is generally widespread in asymptomatic small ruminant flocks. According to the world animal health information system (WAHIS) of OIE, *C. burnetii* is prevalent worldwide, with an exception in New Zealand. Ticks may also play a significant role [13,14]. In Pakistan, Q fever was detected in camels in 1955 for the first time. Limited seroepidemiological data about coxiellosis of sheep and goats exist for Muzaffargarh and Layyah [15]. In Punjab, Q fever is present in Lahore, Sheikhupura, Gujranwala, Faisalabad, DG Khan, Attock, and Sahiwal but was not found in Chakwal and Sargodha in the past [16,17]. 

Besides the landscape and climate, the prevalence of Q fever in animals was also linked to management risk factors, i.e., production system, vaccination, biosafety, and biosecurity of holdings and trading systems. In the rural areas of developing countries, livestock plays and will always play an integral role in the income of the low-income population [18]. About 8 million people are connected to livestock and production activities in Pakistan [17]. These farmers and livestock workers have direct or indirect contact with sheep and goats and these animals are considered the major risk for Q fever infection [4,8,19,20,21]. Neglecting the control of Q fever in these communities at high risk will threaten lives [6]. Additionally, the increasing animal exchange between Asian countries does call for monitoring of the livestock health status to minimize the transboundary spreading of (drug-resistant) pathogens [22]. This seroepidemiological study on the prevalence of *C. burnetii* antibodies in sheep and goat herds of selected districts of Punjab provinces and on potential risk factors will help to develop surveillance and control plans for Pakistan.

## 2. Results

Thirty-four (11.3%) samples were found seropositive. The seroprevalence of Q fever in the selected districts of Kasur, Okara, and Pakpattan was 14%, 9%, and 11%, respectively. The highest *C. burnetii* seropositivity was observed in the district Kasur (*p* > 0.05) (Table. 1). 

The caprine seropositivity for Q fever antibodies (17.1%) was significantly higher when compared to that of sheep (4.9%). The seroprevalence of Q fever was higher in males (12.5%) than in females (11.2%). The seroprevalence of different breeds of sheep and goats were 20.2%, 7.7%, and 4.9% in the Beetle, Teddy, and Kajli breeds, respectively. Rural areas detected a 13.2% positive sera and 9.0% from urban areas which is a significant finding. Prevalence was affected by age and a high prevalence was identified in adult animals (12.5%). ‘Age’ was a statistically significant variable (*p* < 0.045) (Table 1).

The seroprevalence of Q fever was higher (50%) in those animals which had tick infestation as compared to those without (3.2%). The presence of ticks was a significant potential risk factor (*p* = 0.000). Animals with an abortion background had a high prevalence of 12% when compared to animals without. Abortion history did not have a significant impact on the prevalence of Q fever. Animals with fomite contact had the highest seroprevalence (52.3%) when compared to those which had no exposure to fomites (4.3%). Animals that had direct contact with other farm animal species were found to be 48.9% seropositive while the rest (4.7%) had no direct contact with other species. This factor was found to be statistically significant (*p* = 0.00) for the seroprevalence of Q fever. 

Higher seropositivity (13.4% vs. 9.6%) was detected when sharing rams. Remarkably, animals kept in cleaned corals were insignificantly less often seropositive (11.0%) than others (11.9%). No restocked animals brought in were seronegative in contrast to 11.6% self-reared animals. In the transhumance productive system, 23.7% of animals had Q fever antibodies. This was a significant difference from other production systems (*p* < 0.001). Animals on greater farms were more often seropositive (12.3%) as compared to animals of small holdings (9.0%). The health status of animals was a significant factor for Q fever: 50% of emaciated animals were seropositive in contrast to 5.7% of healthy ones (Table 2). 

Animal species, tick infestation, contact with fomites, and contact with other farm animal species were risk factors significantly associated with *C. burnetii* seropositivity on the basis of multivariate analysis (Table 3).

## 3. Discussion

This cross-sectional study on sheep and goats was conducted in various districts of the Punjab Province of Pakistan to determine the prevalence of anti-*C. burnetii* antibodies and to assess associated risk factors to assist in the development of surveillance and control plans. The overall prevalence of antibodies of *C. burnetii* was 11.3%. Our findings are in line with a previous study which found 15.4% small ruminant seropositive for *C. burnetii* in Punjab, Pakistan [10]. However, contrary to these findings a comparatively higher percentage of small ruminants was found seropositive in a prior study in the Punjab region of Pakistan [15]. Climate and landscape may influence the distribution of aerosolized bacteria which is reflected in divergent seroprevalences of narrowly defined areas but may also be true for larger regions. In a nationwide study, 11.4% of small ruminants were positive in Portugal where climate conditions are different from those of the Punjab [11]. In the circumscribed region of the Marmara Sea in Turkey, 13.22% of small ruminants were found to have *C. burnetii* antibodies [4]. The similar values are striking considering the different landscapes and a context to the biology of *C. burnetii* per se can be guessed. Indeed, small variations in *C. burnetii* antibodies prevalence have been recorded for Kasur (14%), Okara (9%), and Pakpattan (11%) in this study. Such variations (4.8% to 65.9%) have also been recorded from small ruminant samples collected at governmental livestock farms located in nine different locations in Pakistan [17]. Similar variations in seroprevalence have been observed in a study that was conducted in small ruminants of the Muzaffargarh and Layyah regions of Pakistani Punjab as well [15]. On the other hand, in a study that was conducted at twelve sites in the Gambia in small ruminants, it was observed that the prevalence of *C. burnetii* can vary greatly from 4% to 47.5% [23]. Variations might be connected to the variations in climatic conditions, humidity, soil texture, and immune system of animals [24,25]. However, a comparison of studies should be performed with great caution as study design, tests used, and other man-made factors may influence the results significantly.

The species-level prevalence was 17.1% and 4.9% in goats and sheep, respectively. There was a significant association (*p* = 0.001) between the prevalence and species of animals based on chi-square analysis. This association was not found in a previous study in sheep (15.6%) and goats (15.0%) in Pakistan [17]. A similar study was conducted in Bangladesh with comparatively higher seroprevalence in sheep (9.52%) and a lower one in goats (3.33%) [26]. Lower seropositivity was recorded in goats (9.3%) and higher in sheep (12.3%) in Northern Ireland [27]. Similarly, as compared to these results, a higher prevalence was documented in sheep (8.9%) and lower (6.8%) in goats in Egypt [1]. Furthermore, researchers from Iran found relatively higher seroprevalences in both sheep (22%) and goats (28.1%) [7]. Most of those owners had no knowledge of zoonotic diseases such as Q fever which is why they did not follow any instructions supporting the management of disease-free farms. Moreover, the large size of those herds also favored the spread of Q fever. The cohabitation of infected small ruminants with other farm animals might also be the reason for disease spread [18,20]. Thus, prevalence studies without international standardization of the experiment set up are not useful for general recommendations but still, they are very helpful to finding local solutions for local problems. 

Male animals are used for breeding purposes and so they are transferred from one area to another and may have contact with many different herds. This method may pose to them a higher risk of acquiring and spreading diseases. Interestingly, males had a higher seroprevalence (12.5%) than females (11.2%) in this study area, a finding that was statistically non-significant. Hence, this finding is not in accordance with previous studies from Pakistan [15], Turkey [6], and various other countries [21]. As Coxiellae have an extraordinary affinity with birth products, fetal membranes, vaginal mucus, and mammary glands, one can suppose that there are simply greater chances of disease spread among females and a higher seroprevalence is normal [1]. A study on domestic ruminants from Bangladesh indeed found more seropositive samples in males (6.67%) than females (4.69%) [26]. 

The breed of the animals plays a role in the prevalence of Q fever in the study area based on Chi-square analysis. However, we acknowledge the fact that the effect of the breed is more likely associated with the species. In our study, we pursued the prevalence of Q fever in one breed of sheep and two breeds of goats. The local breed of goat, i.e., Beetle, had the highest prevalence of *C. burnetii* antibodies. A study in Iran revealed that native breeds of those areas were also more likely to be positive for Q fever (28.9%) [7]. The cohabitation of farm animal species or mixed herds with goats and sharing of water and grazing resources may increase the risk for infection [17]. On the other hand, other regions of the world have lower prevalences of native breeds mated with foreign Boer goats [21]. 

In the present study, most of the samples were collected from animals in rural areas (n = 167; 13.2%) without proper access to facilities for rearing and absence of hygiene regimes, e.g., cleaning of animals and their sheds. These facts may contribute to higher disease prevalences. Thus, not removing contaminated birth products will result in a higher risk for Q fever for men and animals in these rural areas. High sheep density (lambing on meadows or sheep drive) in combination with hot and windy climate conditions resulted in outbreaks in humans in Germany, even close to towns [1,28]. Small ruminants can acquire the disease through environmental contamination via aerosols [19,29]. A statistically significant risk for infection was connected with higher age in this study. This finding is in accordance with those from east Turkey [6], Iran [30,31], and various other countries such as Bangladesh [26], Laos [21], Ethiopia, and Egypt [1,12]. Age simply increases the chance to meet infected animals or a contaminated environment and getting infected.

Ticks are considered an important risk factor and have a significant association with *C. burnetii* seropositivity. In the present study, 50% of seropositive animals were found infested with ticks. Another study from Pakistan revealed that ticks may carry the Q fever pathogen and contaminate the environment or host [17]. This finding is in accordance with findings of a previous study in other regions of Pakistan that found a 97.9% prevalence in those animals which harbored ticks [15]. Research from other countries also documented that infestation with ticks was significantly associated with anti *C. burnetii* antibody prevalence. Various tick species can transfer Coxiellae transstadially as well as transovarially and may spread Q fever from a diseased animal to healthy ones via the blood meal [7,12]. 

It is not surprising that a higher prevalence was observed in those animals which have an abortion history. Various seroepidemiological studies showed that aborted ewes show higher prevalences of *C. burnetii* antibodies [4]. In the Muzaffargarh and Layyah of Pakistan, high prevalences (77.5%) were observed in those animals which had an abortion history also [15]. Soil and the environment can be massively contaminated with *C. burnetii* via abort materials and incorrect disposal of aborted fetuses [12]. On the other hand, previous studies have found that animals shedding the *C. burnetii* apparently had no clear signs and symptoms of disease and had no abortion history [21]. Moreover, seronegative animals can also shed the pathogen without showing any symptoms [19]. The role of fomites is well known for the possible transmission of the Q fever pathogen from one animal to other animals and humans, too [32,33]. Such reports have been documented in Switzerland and Britain, where residents living near roads used for the transport of sheep were infected with Q fever due to contaminated dust and litter [34]. Moreover, abort materials harbor large amounts of pathogens which can be transferred through fomites to surrounding and nearby animals [20]. 

In our study, the contact of animals with other farm animal species was considered a potential risk factor for Q fever which also proved statistically significant (*p* = 0.000). Coxiellae can easily be transferred from one animal to another. In a Dutch study, the presence of animals such as dogs, cats, rabbits, and birds, e.g., chicken, was linked to seropositivity [35].

Rearing rams and sharing them for breeding had no significant impact on the prevalence of Q fever in the current study. Surprisingly, it was observed that artificial insemination was associated with the seropositivity of goats on dairy farms in the Netherlands [35]. Sexual transmission of *C. burnetii* from male animals to female partners was reported from Poland by detection of viable *C. burnetii* in the semen of cattle bulls which were initially found seropositive for Q fever [36]. In another recent study from Russia, researchers also reported PCR-based detection of DNA of *C. burnetii* from semen samples of bulls used for artificial insemination [37]. 

In general, healthy and well-kept animals are more resistant to diseases than poorly kept ones [17,38]. Thus, the management of farms has a significant impact on disease prevalence. Cleaning corrals on a daily basis reduces the chances of the spread of zoonotic diseases such as Q fever in livestock farms. A higher prevalence of *C. burnetii* was observed in sheep and goats in Brazil because these farms were not cleaned on a daily basis [39]. Surprisingly, animals from clean corrals actually had a lower seroprevalence compared to non-clean ones, and regardless the difference was not statistically significant in either the chi-square or multivariate analyses. The reason for this finding is that the time between cleanings was too long as the farmers cleaned the corrals only every fortnight. A similar result was observed in Pakistan for brucellosis in animals housed in such ‘clean’ corrals [18]. As most of these animal samples originated from rural areas, cohabitation and a common lack of good management practices may have fostered the risk of Q fever prevalence. Even good biosafety practices may fail, e.g., pre-testing of newly purchased animals. It has been documented that those animals which had a disease history in the past can act as reservoirs of Q fever even though they exhibit seronegative results [4]. This finding was also strengthened by a study that documented that 24% of seronegative small ruminants shed *C. burnetii* bacteria in the environment [19]. Persistent monitoring and certification of freedom of Q fever need to be an integral part of good livestock farming to guarantee the safe trade of disease-free animals. In some instances, unrestricted access of infected wildlife to farms may also play a role in the cycle of coxiellosis in the Pakistani setting. However, this assumption needs to be proved in the future. Intensive farming practices for commercial purposes promote higher Q fever prevalences because of crowding and frequent cross-species transfer after massive contact of various farm animal species on such farms [21]. 

## 4. Materials and Methods

### 4.1. Sampling Sites

The regions selected for sampling include three administrative subunits of the Punjab province of Pakistan (Figure 1). The district Kasur (31.1165° N, 74.4494° E) is located in the south of district Lahore and the Ganda Singh border. This area is highly fertile land and 61% of the communities earn their livelihood with livestock. The common livestock are sheep, goats, cattle, and buffaloes. According to the 9211 Virtual Governance System, this region has 319,048 small ruminants and 1,060,994 large ruminants. Small ruminant breeds include Teddy (goat = 150,023), Beetal (goat = 92,318) and Kajli (sheep = 7492). Various infectious and zoonotic diseases have been reported in this region [18,40]. Okara district (30.8090° N, 73.4508° E) is located in the east of Kasur. The climate of this region is usually dry and warm, and the temperature of this district varies from 3 °C to 45 °C during the year. Okara is renowned for its agriculture-based economy, cotton mills, and various military dairy farms and the production of cheese from these farms. On the basis of the 9211 Virtual Governance System, breeds of large ruminants (n = 1,312,106) are Sahiwal, Cholistani (cattle), and Nili-Ravi (buffalo). Small ruminant (n = 474,062) breeds include Teddy (goat = 177,744), Beetal (goat = 75,825), and Kajli (sheep = 8841). Prevalent diseases of livestock in this area are foot and mouth disease, milk fever, mastitis, and brucellosis. Fifty-three percent of the human population are engaged in agriculture but only 12% are involved in livestock keeping [18,40]. 

District Pakpattan (30.2527° N, 73.1822° E) is located to the north of the Okara district. The soil of this district is highly fertile and most of the people obtain their income through agriculture, flour mills, oil mills, rice mills, poultry feed production, and textile weaving. Agriculture is the main source of livelihood for 50% of the population while 10% of the population relies on livestock production. According to the 9211 Virtual Governance System, this region has 381,477 small ruminants and 913,574 large animals. The Beetal (n = 57,359) and Teddy (n = 210,472) breed of goats and Kajli (n = 12,799) breed of sheep are considered as the important species of livestock [18,40]. 

### 4.2. Epidemiological Data Collection

The sampling included administrative subunits such as districts and tehsils of Punjab province. The demographic data of the districts are shown in Table 1. Prior to sampling, a predesigned data sheet was filled in to gather demographic and epidemiological data to assess risk factors of Q fever. The owners of the sheep and goats were interviewed in their native language. Geological coordinates of each village were recorded using a GPS receiver. The information about tick infestation was collected by asking owners whether they had taken any precautions against ticks, and by examining the animals and the sheds where they were housed.

### 4.3. Blood Sampling and Serum Separation

A careful literature survey established that in our targeted study areas, no information pertaining to the prevalence of Q fever in small ruminants was present. For calculation of the sample size, a 95% confidence interval with an anticipated prevalence of 50% and preferred absolute precision of 6% was applied [41]. The prerequisite of this process was altogether three hundred serum samples from all farms. Blood samples of 142 sheep (Kalji breed) and 158 goats (Beetal = 119; Teddy = 39) were aseptically collected from their jugular veins. Blood samples were transported to the laboratory at 4 °C. The serum was separated by centrifugation at 3920 g for 5 min. The supernatants were transferred to 1.5 mL Eppendorf tubes with pipettes and stored at −20 °C.

### 4.4. Serological Investigation

A commercially available indirect ELISA test kit for Q fever (IDEXX Laboratories, Liebefeld, Switzerland) was used for the detection of *C. burnetii* specific antibodies. The assay was performed as per the instructions of the manufacturer. Serum was diluted at 1:400 according to recommendations. The specific antibodies were detected by using peroxidase-marked anti-ruminant conjugate. An ELISA reader was used for the measurement of optical densities (OD) of the positive and negative controls and test samples at 450 nm. Results were presented as percentages of the OD reading of the test sample (OD%). The outcomes were interpreted according to the recommendations of the producer, in that S/*p* values ≤ 30% were negative, and values ≥ 40% were positive.

### 4.5. Statistical Analysis

To evaluate statistical significance, the Statistical Package for Social Sciences (SPSS), software version 21.0 was used. The Chi-square test was used to determine the association between risk factors and the categorical variables of sheep and goats from different localities. Binary logistic regression analysis was applied to identify the risk factors associated with seropositive results of Q fever, odds ratios, and the 95% confidence interval levels. A statistically significant result was considered with a *p*-value ≤ 0.05.

## 5. Conclusions

This research proved that Q fever is present in small ruminants in three districts of the Punjab of Pakistan. Serology and direct identification techniques for *Coxiella burnetii* are needed for consistent surveillance of small ruminant production. A safe trading system for animals of disease-free farms has to be implemented to minimize financial losses for the producers and state economy and finally to foster animal welfare and human health.

## Figures and Tables

**Figure 1 pathogens-11-00664-f001:**
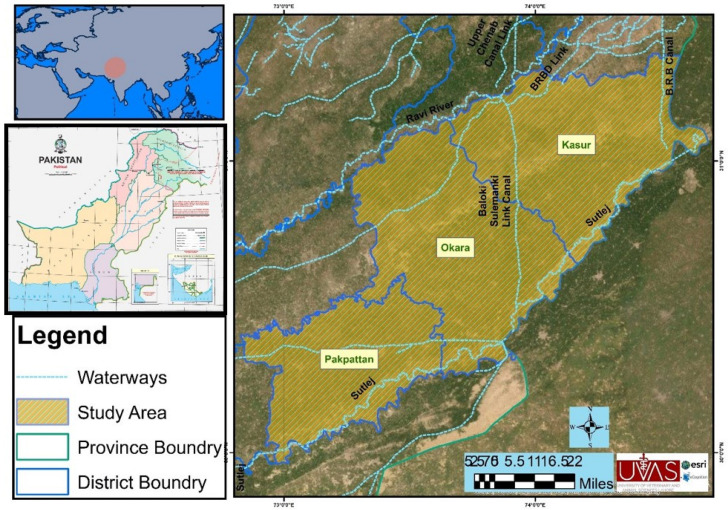
Sampling site of the study area in the eastern region of Punjab, Pakistan.

**Table 1 pathogens-11-00664-t001:** Prevalences of anti-*Coxiella burnetii* antibodies with respect to demographic variables of the sheep and goat populations of three districts of Punjab, Pakistan.

Categories	Variables	Number of Samples	Seroprevalence (%)	Chi-Square	*p*-Value
District	Kasur	100	14(14)	1.266	0.532
	Okara	100	9(9)		
	Pakpattan	100	11(11)		
Species	Sheep	142	7(4.9)	11.003	0.001
	Goats	158	27(17.1)		
Gender	Female	260	29(11.2)	0.063	0.790
	Male	40	5(12.5)		
Breed	Kajli (sheep)	142	7(4.9)	15.552	0.000
	Beetal (goat)	119	24(20.2)		
	Teddy (goat)	39	3(7.7)		
Urbanicity	Rural	167	22(13.2)	1.270	0.278
	Urban	133	12(9)		
Age	Adult	272	34(12.5)	3.947	0.045
	Young	28	0(0)		

**Table 2 pathogens-11-00664-t002:** Risk factors associated with anti-*Coxiella burnetii* antibody prevalence in small ruminants of Punjab, Pakistan based on Chi-square analysis.

Risk Factors	Variables	Seropositive (%)	Examined	Chi-Square	*p*-Value
Ticks Infestation	Yes	26(50)	52	93.59	0.000
	No	8(3.2)	248		
Abortion History	Yes	3(12)	25	0.012	0.558
	No	31(11.3)	275		
Contact with Fomites	Yes	23(52.3)	44	86.00	0.000
	No	11(4.3)	256		
Contact with other Species	Yes	22(48.9)	45	74.036	0.000
	No	12(4.7)	255		
Sharing of Bucks	Yes	18(13.4)	134	1.062	0.198
	No	16(9.6)	166		
Cleaning of the Corrals	Yes	22(11.0)	199	0.045	0.485
	No	12(11.9)	101		
Stock Replacement	Purchased	0(0)	7	0.916	0.427
	Reared	34(11.6)	293		
Production System	Sedentary	25(9.5)	262	6.605	0.016
	Transhumant	9(23.7)	38		
Farm Type	Small holder	8(9.0)	89	0.692	0.268
	Commercial	26(12.3)	211		
Health Status	Healthy	15(5.7)	262	64.738	0.000
	Emaciated	19(50)	38		

**Table 3 pathogens-11-00664-t003:** Risk factors associated with anti-*Coxiella burnetii* antibody prevalence logistic regression analysis model.

Variables	*p*-Value	OD *	(95% C.I *)	
			Lower	Upper
District	0.885	4.37 × 10^20^	0	4.23 × 10^300^
Animal Species	0.028	0	0	0.396
Gender	0.17	0.009	0	7.384
Breed	0.091	104.334	0.474	22,985.8
Urbanicity	0.97	0.951	0.07	12.893
Age	0.998	11,802,766	0	
Ticks Infestation	0.003	248.035	6.711	9167.067
Abortion History	0.874	0.758	0.025	23.212
Contact with Fomites	0.034	25.397	1.273	506.886
Contact with other Species	0.006	43.564	3.007	631.097
Sharing of Bucks	0.233	54.946	0.076	39,650.9
Cleaning of the Corrals	0.144	10.431	0.449	242.22
Stock Replacement	0.999	0	0	
Production System	0.189	0.125	0.006	2.783
Farm Type	0.227	0.213	0.017	2.609
Health Status	0.739	0.552	0.017	18.239

* OD = Odd ratio, C.I = Confidence interval.

## Data Availability

The data presented in this study are available within the article.

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
