# Peer review of "Sero-Epidemiology of Coxiella burnetii Infection in Small Ruminants in the Eastern Region of Punjab, Pakistan"

_pathogens, 2022, doi:10.3390/pathogens11060664_

Round 1

Reviewer 1 Report

The authors present a study of the seroprevalence of Coxiella burnetii (Q fever) in sheep and goats in the Punjab of Pakistan.  They report a higher prevalence in goats and show some variables that are significant in a multivariate analysis.  The study is both useful for knowing the exposure to Q fever in the region as well as helping determine which factors may be contributing to its spread.  In the process of doing so, they also provide a well-researched context for their findings in the Introduction and Discussion sections.  I have suggested revisions in two categories: ones I consider essential for accuracy and readability, and optional ones that would help the paper but do not need to be made. 

Essential revisions:

Lines 42-43: “109 bacteria per gram” should be an exponent, e.g. 10 to the ninth power (109).  In addition, please be advised that the citation for this study is incorrect.  While the study cited does mention this fact, it did not come from that study but from Babudieri 1959 Adv. Vet. Sci

Lines 56-58: Do you have a source for the signs of Q fever in animals described?  References 11 and 12 are serology studies and do not describe signs except in their introduction.  I know of reproductive syndromes, but I have not heard of Q fever causing genital disorders, arthritis, or mastitis in animals, and neither of the papers referenced mentions these particular symptoms.  References should be provided for these signs, or they should be removed if listed in error.

Line 135: There may be a typo here.  I believe it should read “conditions are different” instead of “conditions a different”.

Lines 168-174: The Discussion states that females had a higher seroprevalence than males in this study, but the Results and Table 1 state males had a higher prevalence than females.  Please align these to whatever the correct amount is, and if necessary update all of the following statements.  I would also not write as if one sex had a higher prevalence than the other when the difference was found to be non-significant.   

Lines 177-178: I do not see an indication from the data that breed plays a significant role in the prevalence of Q fever in the study area.  The breed analysis was done pooling data from sheep and goats, meaning that the chi-square test was not only testing for the effect of breed but from species.  Because we know species has a significant effect in its own chi-square analysis, one cannot rule out that it was in fact species and not breed that made “breed” significant.  This hypothesis is more consistent with the results of the multivariate analysis, which found species but not breed as a significant factor.  In addition, only one breed of sheep (Kajli) was tested as compared to two breeds of goat (Beetal and Teddy).  I would request this be addressed, either by acknowledging that the effect of breed appears to be due to species and mentioning only one breed of sheep was tested, by removing breed from the analysis, or (I do not think this last one is necessary) by re-doing some or all analyses separately by species.  While I personally agree that breed has been a significant factor in other studies and suspect host genetics play a role in Q fever resistance in general, there is not any indication that it plays a role in this study population. 

Lines 238-240: I would revise the statement about clean corrals.  According to Table 1, animals from clean corrals actually had a lower seroprevalence compared to non-clean ones, and regardless the difference was not statistically significant in either the chi-square or multivariate analyses.  I think the following statement about the time between cleanings still makes sense, but I would treat the data as if there were no difference, as statistically there was not.

Optional revisions:

Line 20: The “n” designation is used, at least in my experience, for an absolute number, with the percentage coming after.  For example “A significant number of the 50 animals surveyed (n=10; 20%) were infected”. 

Line 110: Beginning the sentence with parentheses is not necessary and is a little confusing.  IT could be “0% of restocked animals”, or simply “No restocked animals”.

Lines 181-183: The effect of cohabitation is interesting, and I suspect the real variable at play here is exposure to goats, which have a higher seroprevalence than sheep in many studies (see McQuiston and Childs 2002, Vector Borne Zoonotic Dis. if interested, which shows a wide difference in the United States).  This area could be developed in the Discussion if you would like, but it is in no way necessary to the paper to do so.

Author Response

Red color text

Comment 1:

The authors present a study of the seroprevalence of Coxiella burnetii (Q fever) in sheep and goats in the Punjab of Pakistan.  They report a higher prevalence in goats and show some variables that are significant in a multivariate analysis.  The study is both useful for knowing the exposure to Q fever in the region as well as helping determine which factors may be contributing to its spread.  In the process of doing so, they also provide a well-researched context for their findings in the Introduction and Discussion sections.  I have suggested revisions in two categories: ones I consider essential for accuracy and readability, and optional ones that would help the paper but do not need to be made. 

Response 1: we are grateful to reviewer for such nice words for our paper and kind suggestions for improvement of paper

 Comment 2:

Essential revisions:

Lines 42-43: “109 bacteria per gram” should be an exponent, e.g. 10 to the ninth power (109).  In addition, please be advised that the citation for this study is incorrect.  While the study cited does mention this fact, it did not come from that study but from Babudieri 1959 Adv. Vet. Sci

Response 2: corrected now

Comment 3:

Lines 56-58: Do you have a source for the signs of Q fever in animals described?  References 11 and 12 are serology studies and do not describe signs except in their introduction.  I know of reproductive syndromes, but I have not heard of Q fever causing genital disorders, arthritis, or mastitis in animals, and neither of the papers referenced mentions these particular symptoms.  References should be provided for these signs, or they should be removed if listed in error.

Response 3: sentence has been modified and appropriate references added now

Comment 4:

Line 135: There may be a typo here.  I believe it should read “conditions are different” instead of “conditions a different”.

Response 4: corrected now

Comment 5:

Lines 168-174: The Discussion states that females had a higher seroprevalence than males in this study, but the Results and Table 1 state males had a higher prevalence than females.  Please align these to whatever the correct amount is, and if necessary update all of the following statements.  I would also not write as if one sex had a higher prevalence than the other when the difference was found to be non-significant. 

Response 5:  discussion modified and aligned as per findings of Table 1 and results.

Comment 6:

Lines 177-178: I do not see an indication from the data that breed plays a significant role in the prevalence of Q fever in the study area.  The breed analysis was done pooling data from sheep and goats, meaning that the chi-square test was not only testing for the effect of breed but from species.  Because we know species has a significant effect in its own chi-square analysis, one cannot rule out that it was in fact species and not breed that made “breed” significant.  This hypothesis is more consistent with the results of the multivariate analysis, which found species but not breed as a significant factor.  In addition, only one breed of sheep (Kajli) was tested as compared to two breeds of goat (Beetal and Teddy).  I would request this be addressed, either by acknowledging that the effect of breed appears to be due to species and mentioning only one breed of sheep was tested, by removing breed from the analysis, or (I do not think this last one is necessary) by re-doing some or all analyses separately by species.  While I personally agree that breed has been a significant factor in other studies and suspect host genetics play a role in Q fever resistance in general, there is not any indication that it plays a role in this study population. 

Response 6: sentences have modified and added as per suggestion. Moreover, role of species also mentioned, line 178-181 now

Comment 7:

Lines 238-240: I would revise the statement about clean corrals.  According to Table 1, animals from clean corrals actually had a lower seroprevalence compared to non-clean ones, and regardless the difference was not statistically significant in either the chi-square or multivariate analyses.  I think the following statement about the time between cleanings still makes sense, but I would treat the data as if there were no difference, as statistically there was not.

Response 7: we are agree with reviewer suggestion and suggested sentences have been added now line 243-245. However, other sentences provided very useful information’s that’s why they are still part of this manuscript

Optional revisions:

Comment 8:

Line 20: The “n” designation is used, at least in my experience, for an absolute number, with the percentage coming after.  For example “A significant number of the 50 animals surveyed (n=10; 20%) were infected”. 

Response 8: we are agree with reviewer comment. Statement is corrected now

Comment 9:

Line 110: Beginning the sentence with parentheses is not necessary and is a little confusing.  IT could be “0% of restocked animals”, or simply “No restocked animals”.

Response 9: corrected now

Comment 10:

Lines 181-183: The effect of cohabitation is interesting, and I suspect the real variable at play here is exposure to goats, which have a higher seroprevalence than sheep in many studies (see McQuiston and Childs 2002, Vector Borne Zoonotic Dis. if interested, which shows a wide difference in the United States).  This area could be developed in the Discussion if you would like, but it is in no way necessary to the paper to do so.

Response 10: we are grateful to reviewer suggestion. But we will consider this reference for some other paper. Regards

Reviewer 2 Report

The study is interesting but there are many imperfections in the drafting of the document.
The experimental work is interesting but there was no sampling carried out according to the expected prevalence.
It is not recommended to mix sheep and goats because they do not have the same sensitivity and if they are separated, the number of individuals is too low.
The number of herds is not indicated nor the prevalence of herds. The age classes are not indicated nor the size of the herds.
The statistical study contains inconsistencies, the confidence intervals are very wide.

Author Response

Yellow color text

The study is interesting but there are many imperfections in the drafting of the document.

Comment 1:
The experimental work is interesting but there was no sampling carried out according to the expected prevalence.

Response 1: required information is added now, line 295-299

Comment 2:
It is not recommended to mix sheep and goats because they do not have the same sensitivity and if they are separated, the number of individuals is too low.

Response 2: very are grateful for reviewer comment. However, in Pakistan sheep and goats are kept in the form of mix at small hold and large farm. That’s why we include both in study.

comment 3:

The number of herds is not indicated nor the prevalence of herds. The age classes are not indicated nor the size of the herds.

Response 3: The purpose of this study was determination of C. burnetii antibodies in small ruminants at individual level that’s why no information of the herd was added. However, age was distribution was given as young and adult which more convenient approach

Comment 4:

The statistical study contains inconsistencies, the confidence intervals are very wide.

Response 4: During statistical analysis, initially we used the chi-square test as a univariate test. Apart from significant variables in chi-square analysis, we include non-significant variables in regression analysis. That is why these non-significant variables have such a wide confidence interval but most of the significant variables have normal CI.

Reviewer 3 Report

Although previous reports have already been published about Q fever infections in small ruminants in Punjab, the present paper could be useful to further stress out the importance of surveillance in farms to prevent potential zoonotic outbreaks.     

Nevertheless, these results would be more interesting and accurate if associated with molecular epidemiology. Could Authors be able to provide such data from the same target population?

As for the text, please check the following sentences:

  • Line 65: WAHIS actually shows a greater number of Countries where burnetii is present. Please update the information;
  • Line 135: it should probably say “… climate conditions ARE different…”;
  • Lines 158-159: as for other Asian countries, previous results showed greater prevalence rates in small ruminants, especially higher in sheep (e.g. from Lebanon [Dabaya et al, 2019], where sheep also resulted as important risk factors when present in mixed heards). Any explanations about possible reasons for such different findings and for the significantly higher goat positivity in your study would be interesting.

Furthermore, the text would be clearer if Authors follow the usual order between "Materials and methods" and Results" section. 

Author Response

Green color text

Comment 1:

Although previous reports have already been published about Q fever infections in small ruminants in Punjab, the present paper could be useful to further stress out the importance of surveillance in farms to prevent potential zoonotic outbreaks.     

Nevertheless, these results would be more interesting and accurate if associated with molecular epidemiology. Could Authors be able to provide such data from the same target population?

Response 1: We are grateful for the kind words of the reviewer. The main purpose of this study was determination of Seroepidemiology of Q fever in small ruminants. We did not use a molecular approach. However, we will use it in future studies

As for the text, please check the following sentences:

Comment 2:

  • Line 65: WAHIS actually shows a greater number of Countries where burnetiiis present. Please update the information;

Response 2: information is updated now

Comment 3:

Line 135: it should probably say “… climate conditions ARE different…”;

Response 3: already corrected as per suggestion of reviewer 1

Comment 4:

Lines 158-159: as for other Asian countries, previous results showed greater prevalence rates in small ruminants, especially higher in sheep (e.g. from Lebanon [Dabaya et al, 2019], where sheep also resulted as important risk factors when present in mixed heards). Any explanations about possible reasons for such different findings and for the significantly higher goat positivity in your study would be interesting.

Response 4: Thank you for your kind suggestions. However we have already compared our results with multiple countries. Possible reason of variation in prevalence of Q fever in small ruminants of different countries also given.

Comment 5:

Furthermore, the text would be clearer if Authors follow the usual order between "Materials and methods" and Results" section.

Response 5:
Thank you for your suggestions. However, this is the standard format provided by the journal in its template.

Round 2

Reviewer 2 Report

It would be useful for other studies to be much more rigorous in the sampling (number of animals chosen according to the expected prevalence) by avoiding the mixing of animal species and also by improving the statistical analysis.

Author Response

Comment 1:

It would be useful for other studies to be much more rigorous in the sampling (number of animals chosen according to the expected prevalence) by avoiding the mixing of animal species and also by improving the statistical analysis.

Response 1: We are grateful to the reviewer for kind suggestions. We will keep in mind these suggestions for future studies
